# Task-based End-to-end Model Learning in Stochastic Optimization

**Priya L. Donti**
Dept. of Computer Science
Dept. of Engr. & Public Policy
Carnegie Mellon University
Pittsburgh, PA 15213
pdonti@cs.cmu.edu

**Brandon Amos**
Dept. of Computer Science
Carnegie Mellon University
Pittsburgh, PA 15213
bamos@cs.cmu.edu

**J. Zico Kolter**
Dept. of Computer Science
Carnegie Mellon University
Pittsburgh, PA 15213
zkolter@cs.cmu.edu

## Abstract

With the increasing popularity of machine learning techniques, it has become common to see prediction algorithms operating within some larger process. However, the criteria by which we train these algorithms often differ from the ultimate criteria on which we evaluate them. This paper proposes an end-to-end approach for learning probabilistic machine learning models in a manner that directly captures the ultimate task-based objective for which they will be used, within the context of stochastic programming. We present three experimental evaluations of the proposed approach: a classical inventory stock problem, a real-world electrical grid scheduling task, and a real-world energy storage arbitrage task. We show that the proposed approach can outperform both traditional modeling and purely black-box policy optimization approaches in these applications.

## 1 Introduction

While prediction algorithms commonly operate within some larger process, the criteria by which we train these algorithms often differ from the ultimate criteria on which we evaluate them: the performance of the full "closed-loop" system on the ultimate task at hand. For instance, instead of merely classifying images in a standalone setting, one may want to use these classifications within planning and control tasks such as autonomous driving. While a typical image classification algorithm might optimize accuracy or log likelihood, in a driving task we may ultimately care more about the difference between classifying a pedestrian as a tree vs. classifying a garbage can as a tree. Similarly, when we use a probabilistic prediction algorithm to generate forecasts of upcoming electricity demand, we then want to use these forecasts to minimize the costs of a scheduling procedure that allocates generation for a power grid. As these examples suggest, instead of using a "generic loss," we instead may want to learn a model that approximates the ultimate task-based "true loss."

This paper considers an end-to-end approach for learning probabilistic machine learning models that directly capture the objective of their ultimate task. Formally, we consider probabilistic models in the context of stochastic programming, where the goal is to minimize some expected cost over the models' probabilistic predictions, subject to some (potentially also probabilistic) constraints. As mentioned above, it is common to approach these problems in a two-step fashion: first to fit a predictive model to observed data by minimizing some criterion such as negative log-likelihood, and then to use this model to compute or approximate the necessary expected costs in the stochastic programming setting. While this procedure can work well in many instances, it ignores the fact that the true cost of the system (the optimization objective evaluated on *actual* instantiations in the real world) may benefit from a model that actually attains worse overall likelihood, but makes more accurate predictions over certain manifolds of the underlying space.

We propose to train a probabilistic model not (solely) for predictive accuracy, but so that–when it is later used within the loop of a stochastic programming procedure–it produces solutions that minimize the ultimate task-based loss. This formulation may seem somewhat counterintuitive, given that a "perfect" predictive model would of course also be the optimal model to use within a stochastic programming framework. However, the reality that all models *do* make errors illustrates that we should indeed look to a final task-based objective to determine the proper error tradeoffs within a machine learning setting. This paper proposes one way to evaluate task-based tradeoffs in a fully automated fashion, by computing derivatives through the solution to the stochastic programming problem in a manner that can improve the underlying model.

We begin by presenting background material and related work in areas spanning stochastic programming, end-to-end training, and optimizing alternative loss functions. We then describe our approach within the formal context of stochastic programming, and give a generic method for propagating task loss through these problems in a manner that can update the models. We report on three experimental evaluations of the proposed approach: a classical inventory stock problem, a real-world electrical grid scheduling task, and a real-world energy storage arbitrage task. We show that the proposed approach outperforms traditional modeling and purely black-box policy optimization approaches.

## 2   Background and related work

**Stochastic programming**    Stochastic programming is a method for making decisions under uncertainty by modeling or optimizing objectives governed by a random process. It has applications in many domains such as energy [1], finance [2], and manufacturing [3], where the underlying probability distributions are either known or can be estimated. Common considerations include how to best model or approximate the underlying random variable, how to solve the resulting optimization problem, and how to then assess the quality of the resulting (approximate) solution [4].

In cases where the underlying probability distribution is known but the objective cannot be solved analytically, it is common to use Monte Carlo sample average approximation methods, which draw multiple iid samples from the underlying probability distribution and then use deterministic optimization methods to solve the resultant problems [5]. In cases where the underlying distribution is not known, it is common to learn or estimate some model from observed samples [6].

**End-to-end training**    Recent years have seen a dramatic increase in the number of systems building on so-called "end-to-end" learning. Generally speaking, this term refers to systems where the end goal of the machine learning process is directly predicted from raw inputs [e.g. 7, 8]. In the context of deep learning systems, the term now traditionally refers to architectures where, for example, there is no explicit encoding of hand-tuned features on the data, but the system directly predicts what the image, text, etc. is from the raw inputs [9, 10, 11, 12, 13]. The context in which we use the term end-to-end is similar, but slightly more in line with its older usage: instead of (just) attempting to learn an output (with known and typically straightforward loss functions), we are specifically attempting to learn a model based upon an end-to-end *task* that the user is ultimately trying to accomplish. We feel that this concept–of describing the entire closed-loop performance of the system as evaluated on the real task at hand–is beneficial to add to the notion of end-to-end learning.

Also highly related to our work are recent efforts in end-to-end policy learning [14], using value iteration effectively as an optimization procedure in similar networks [15], and multi-objective optimization [16, 17, 18, 19]. These lines of work fit more with the "pure" end-to-end approach we discuss later on (where models are eschewed for pure function approximation methods), but conceptually the approaches have similar motivations in modifying typically-optimized policies to address some task(s) directly. Of course, the actual methodological approaches are quite different, given our specific focus on stochastic programming as the black box of interest in our setting.

**Optimizing alternative loss functions**    There has been a great deal of work in recent years on using machine learning procedures to optimize different loss criteria than those "naturally" optimized by the algorithm. For example, Stoyanov et al. [20] and Hazan et al. [21] propose methods for optimizing loss criteria in structured prediction that are *different* from the inference procedure of the prediction algorithm; this work has also recently been extended to deep networks [22]. Recent work has also explored using auxiliary prediction losses to satisfy multiple objectives [23], learning

dynamics models that maximize control performance in Bayesian optimization [24], and learning adaptive predictive models via differentiation through a meta-learning optimization objective [25].

The work we have found in the literature that most closely resembles our approach is the work of Bengio [26], which uses a neural network model for predicting financial prices, and then optimizes the model based on returns obtained via a hedging strategy that employs it. We view this approach–of both using a model and then tuning that model to adapt to a (differentiable) procedure–as a philosophical predecessor to our own work. In concurrent work, Elmachtoub and Grigas [27] also propose an approach for tuning model parameters given optimization results, but in the context of linear programming and outside the context of deep networks. Whereas Bengio [26] and Elmachtoub and Grigas [27] use hand-crafted (but differentiable) algorithms to approximately attain some objective given a predictive model, our approach is tightly coupled to stochastic programming, where the explicit objective is to *attempt* to optimize the desired task cost via an exact optimization routine, but given underlying randomness. The notions of stochasticity are thus naturally quite different in our work, but we do hope that our work can bring back the original idea of task-based model learning. (Despite Bengio [26]'s original paper being nearly 20 years old, virtually all follow-on work has focused on the financial application, and not on what we feel is the core idea of using a surrogate model within a task-driven optimization procedure.)

## 3 End-to-end model learning in stochastic programming

We first formally define the stochastic modeling and optimization problems with which we are concerned. Let $(x \in \mathcal{X}, y \in \mathcal{Y}) \sim \mathcal{D}$ denote standard input-output pairs drawn from some (real, unknown) distribution $\mathcal{D}$. We also consider actions $z \in \mathcal{Z}$ that incur some expected loss $L_{\mathcal{D}}(z) = E_{x,y \sim \mathcal{D}}[f(x, y, z)]$. For instance, a power systems operator may try to allocate power generators $z$ given past electricity demand $x$ and future electricity demand $y$; this allocation's loss corresponds to the over- or under-generation penalties incurred given future demand instantiations.

If we knew $\mathcal{D}$, then we could select optimal actions $z_{\mathcal{D}}^{\star} = \operatorname{argmin}_z L_{\mathcal{D}}(z)$. However, in practice, the true distribution $\mathcal{D}$ is unknown. In this paper, we are interested in modeling the conditional distribution $y|x$ using some parameterized model $p(y|x; \theta)$ in order to minimize the real-world cost of the policy implied by this parameterization. Specifically, we find some parameters $\theta$ to parameterize $p(y|x; \theta)$ (as in the standard statistical setting) and then determine optimal actions $z^{\star}(x; \theta)$ (via stochastic optimization) that correspond to our observed input $x$ and the specific choice of parameters $\theta$ in our probabilistic model. Upon observing the costs of these actions $z^{\star}(x; \theta)$ relative to true instantiations of $x$ and $y$, we update our parameterized model $p(y|x; \theta)$ accordingly, calculate the resultant new $z^{\star}(x; \theta)$, and repeat. The goal is to find parameters $\theta$ such that the corresponding policy $z^{\star}(x; \theta)$ optimizes the loss under the *true* joint distribution of $x$ and $y$.

Explicitly, we wish to choose $\theta$ to minimize the *task loss* $L(\theta)$ in the context of $x, y \sim \mathcal{D}$, i.e.

$$\underset{\theta}{\text{minimize}} \quad L(\theta) = \mathbf{E}_{x,y \sim \mathcal{D}}[f(x, y, z^{\star}(x; \theta))]. \tag{1}$$

Since in reality we do not know the distribution $\mathcal{D}$, we obtain $z^{\star}(x; \theta)$ via a proxy stochastic optimization problem for a fixed instantiation of parameters $\theta$, i.e.

$$z^{\star}(x; \theta) = \underset{z}{\text{argmin}} \quad \mathbf{E}_{y \sim p(y|x; \theta)}[f(x, y, z)]. \tag{2}$$

The above setting specifies $z^{\star}(x; \theta)$ using a simple (unconstrained) stochastic program, but in reality our decision may be subject to both probabilistic and deterministic constraints. We therefore consider more general decisions produced through a generic stochastic programming problem[1]

$$
\begin{aligned}
z^{\star}(x; \theta) = \underset{z}{\text{argmin}} \quad & \mathbf{E}_{y \sim p(y|x; \theta)}[f(x, y, z)] \\
\text{subject to} \quad & \mathbf{E}_{y \sim p(y|x; \theta)}[g_i(x, y, z)] \leq 0, \quad i = 1, \ldots, n_{ineq} \\
& h_i(z) = 0, \quad i = 1, \ldots, n_{eq}.
\end{aligned} \tag{3}
$$

In this setting, the full task loss is more complex, since it captures both the expected cost and any deviations from the constraints. We can write this, for instance, as

$$L(\theta) = \mathbf{E}_{x,y\sim\mathcal{D}}[f(x,y,z^\star(x;\theta))] + \sum_{i=1}^{n_{ineq}} I\{\mathbf{E}_{x,y\sim\mathcal{D}}[g_i(x,y,z^\star(x;\theta))] \leq 0\} + \sum_{i=1}^{n_{eq}} \mathbf{E}_x[I\{h_i(z^\star(x;\theta)) = 0\}]$$

(4)

(where $I(\cdot)$ is the indicator function that is zero when its constraints are satisfied and infinite otherwise). However, the basic intuition behind our approach remains the same for both the constrained and unconstrained cases: in both settings, we attempt to learn parameters of a probabilistic model not to produce strictly "accurate" predictions, but such that *when we use the resultant model within a stochastic programming setting, the resulting decisions perform well under the true distribution.*

Actually solving this problem requires that we differentiate through the "argmin" operator $z^\star(x;\theta)$ of the stochastic programming problem. This differentiation is not possible for all classes of optimization problems (the argmin operator may be discontinuous), but as we will show shortly, in many practical cases–including cases where the function and constraints are strongly convex–we can indeed efficiently compute these gradients even in the context of constrained optimization.

## 3.1 Discussion and alternative approaches

We highlight our approach in contrast to two alternative existing methods: traditional model learning and model-free black-box policy optimization. In traditional machine learning approaches, it is common to use $\theta$ to minimize the (conditional) log-likelihood of observed data under the model $p(y|x;\theta)$. This method corresponds to approximately solving the optimization problem

$$\underset{\theta}{\text{minimize}} \quad \mathbf{E}_{x,y\sim\mathcal{D}}\left[-\log p(y|x;\theta)\right].$$

(5)

If we then need to use the conditional distribution $y|x$ to determine actions $z$ within some later optimization setting, we commonly use the predictive model obtained from (5) directly. This approach has obvious advantages, in that the model-learning phase is well-justified independent of any future use in a task. However, it is also prone to poor performance in the common setting where the true distribution $y|x$ cannot be represented within the class of distributions parameterized by $\theta$, i.e. where the procedure suffers from model bias. Conceptually, the log-likelihood objective *implicitly* trades off between model error in different regions of the input/output space, but does so in a manner largely opaque to the modeler, and may ultimately *not* employ the correct tradeoffs for a given task.

In contrast, there is an alternative approach to solving (1) that we describe as the model-free "black-box" policy optimization approach. Here, we forgo learning any model at all of the random variable $y$. Instead, we attempt to learn a policy mapping directly from inputs $x$ to actions $z^\star(x;\bar{\theta})$ that minimize the loss $L(\bar{\theta})$ presented in (4) (where here $\bar{\theta}$ defines the form of the policy itself, not a predictive model). While such model-free methods can perform well in many settings, they are often very data-inefficient, as the policy class must have enough representational power to describe sufficiently complex policies without recourse to any underlying model.[2]

**Algorithm 1** Task Loss Optimization

1: **input:** $\mathcal{D}$    *// samples from true distribution*
2: **initialize** $\theta$   *// some initial parameterization*

3: **for** $t = 1, \ldots, T$ **do**
4:    **sample** $(x, y) \sim \mathcal{D}$
5:    **compute** $z^\star(x;\theta)$ via Equation (3)

6:    *// step in violated constraint or objective*
7:    **if** $\exists i$ s.t. $g_i(x,y,z^\star(x;\theta)) > 0$ **then**
8:      **update** $\theta$ with $\nabla_\theta g_i(x,y,z^\star(x;\theta))$
9:    **else**
10:      **update** $\theta$ with $\nabla_\theta f(x,y,z^\star(x;\theta))$
11:    **end if**
12: **end for**

Our approach offers an intermediate setting, where we *do* still use a surrogate model to determine an optimal decision $z^\star(x;\theta)$, yet we adapt this model based on the task loss instead of any model prediction accuracy. In practice, we typically want to minimize some weighted combination of log-likelihood *and* task loss, which can be easily accomplished given our approach.

## 3.2 Optimizing task loss

To solve the generic optimization problem (4), we can in principle adopt a straightforward (constrained) stochastic gradient approach, as detailed in Algorithm 1. At each iteration, we

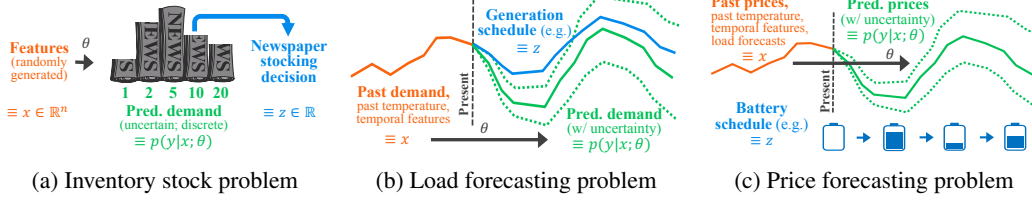

<p style="text-align:center">(a) Inventory stock problem      (b) Load forecasting problem      (c) Price forecasting problem</p>

Figure 1: Features $x$, model predictions $y$, and policy $z$ for the three experiments.

solve the proxy stochastic programming problem (3) to obtain $z^\star(x, \theta)$, using the distribution defined by our current values of $\theta$. Then, we compute the true loss $L(\theta)$ using the observed value of $y$. If any of the inequality constraints $g_i$ in $L(\theta)$ are violated, we take a gradient step in the violated constraint; otherwise, we take a gradient step in the optimization objective $f$. We note that if any inequality constraints are probabilistic, Algorithm 1 must be adapted to employ mini-batches in order to determine whether these probabilistic constraints are satisfied. Alternatively, because even the $g_i$ constraints are probabilistic, it is common in practice to simply move a weighted version of these constraints to the objective, i.e., we modify the objective by adding some appropriate penalty times the positive part of the function, $\lambda g_i(x, y, z)_+$, for some $\lambda > 0$. In practice, this has the effect of taking gradient steps jointly in all the violated constraints and the objective in the case that one or more inequality constraints are violated, often resulting in faster convergence. Note that we need only move stochastic constraints into the objective; deterministic constraints on the policy itself will always be satisfied by the optimizer, as they are independent of the model.

### 3.3 Differentiating the optimization solution to a stochastic programming problem

While the above presentation highlights the simplicity of the proposed approach, it avoids the issue of chief technical challenge to this approach, which is computing the gradient of an objective that depends upon the argmin operation $z^\star(x; \theta)$. Specifically, we need to compute the term

$$\frac{\partial L}{\partial \theta} = \frac{\partial L}{\partial z^\star} \frac{\partial z^\star}{\partial \theta} \tag{6}$$

which involves the Jacobian $\frac{\partial z^\star}{\partial \theta}$. This is the Jacobian of the optimal solution with respect to the distribution parameters $\theta$. Recent approaches have looked into similar argmin differentiations [28, 29], though the methodology we present here is more general and handles the stochasticity of the objective.

At a high level, we begin by writing the KKT optimality conditions of the general stochastic programming problem (3). Differentiating these equations and applying the implicit function theorem gives a set of linear equations that we can solve to obtain the necessary Jacobians (with expectations over the distribution $y \sim p(y|x; \theta)$ denoted $\mathbf{E}_{y_\theta}$, and where $g$ is the vector of inequality constraints)

$$\begin{bmatrix} \nabla_z^2 \mathbf{E}_{y_\theta} f(z) + \sum_{i=1}^{n_{ineq}} \lambda_i \nabla_z^2 \mathbf{E}_{y_\theta} g_i(z) & (\nabla_z \mathbf{E}_{y_\theta} g(z))^T & A^T \\ \mathrm{diag}(\lambda)\left(\nabla_z \mathbf{E}_{y_\theta} g(z)\right) & \mathrm{diag}(\mathbf{E}_{y_\theta} g(z)) & 0 \\ A & 0 & 0 \end{bmatrix} \begin{bmatrix} \frac{\partial z}{\partial \theta} \\ \frac{\partial \lambda}{\partial \theta} \\ \frac{\partial \nu}{\partial \theta} \end{bmatrix} = \begin{bmatrix} \frac{\partial \nabla_z \mathbf{E}_{y_\theta} f(z)}{\partial \theta} + \frac{\partial \sum_{i=1}^{n_{ineq}} \lambda_i \nabla_z \mathbf{E}_{y_\theta} g_i(z)}{\partial \theta} \\ \mathrm{diag}(\lambda) \frac{\partial \mathbf{E}_{y_\theta} g(z)}{\partial \theta} \\ 0 \end{bmatrix}. \tag{7}$$

The terms in these equations look somewhat complex, but fundamentally, the left side gives the optimality conditions of the convex problem, and the right side gives the derivatives of the relevant functions at the achieved solution with respect to the governing parameter $\theta$. In practice, we calculate the right-hand terms by employing sequential quadratic programming [30] to find the optimal policy $z^\star(x; \theta)$ for the given parameters $\theta$, using a recently-proposed approach for fast solution of the argmin differentiation for QPs [31] to solve the necessary linear equations; we then take the derivatives at the optimum produced by this strategy. Details of this approach are described in the appendix.

## 4 Experiments

We consider three applications of our task-based method: a synthetic inventory stock problem, a real-world energy scheduling task, and a real-world battery arbitrage task. We demonstrate that the task-based end-to-end approach can substantially improve upon other alternatives. Source code for all experiments is available at `https://github.com/locuslab/e2e-model-learning`.

### 4.1 Inventory stock problem

**Problem definition** To highlight the performance of the algorithm in a setting where the true underlying model is known to us, we consider a "conditional" variation of the classical inventory stock problem [4]. In this problem, a company must order some quantity $z$ of a product to minimize costs over some stochastic demand $y$, whose distribution in turn is affected by some observed features $x$ (Figure 1a). There are linear and quadratic costs on the amount of product ordered, plus different linear/quadratic costs on over-orders $[z - y]_+$ and under-orders $[y - z]_+$. The objective is given by

$$f_{stock}(y, z) = c_0 z + \frac{1}{2} q_0 z^2 + c_b [y - z]_+ + \frac{1}{2} q_b ([y - z]_+)^2 + c_h [z - y]_+ + \frac{1}{2} q_h ([z - y]_+)^2, \quad (8)$$

where $[v]_+ \equiv \max\{v, 0\}$. For a specific choice of probability model $p(y|x; \theta)$, our proxy stochastic programming problem can then be written as

$$\underset{z}{\text{minimize}} \quad \mathbf{E}_{y \sim p(y|x;\theta)}[f_{stock}(y, z)]. \quad (9)$$

To simplify the setting, we further assume that the demands are discrete, taking on values $d_1, \ldots, d_k$ with probabilities (conditional on $x$) $(p_\theta)_i \equiv p(y = d_i | x; \theta)$. Thus our stochastic programming problem (9) can be written succinctly as a joint quadratic program[3]

$$\underset{z \in \mathbb{R}, z_b, z_h \in \mathbb{R}^k}{\text{minimize}} \quad c_0 z + \frac{1}{2} q_0 z^2 + \sum_{i=1}^{k} (p_\theta)_i \left( c_b (z_b)_i + \frac{1}{2} q_b (z_b)_i^2 + c_h (z_h)_i + \frac{1}{2} q_h (z_h)_i^2 \right) \quad (10)$$

$$\text{subject to} \quad d - z\mathbf{1} \leq z_b, \quad z\mathbf{1} - d \leq z_h, \quad z, z_h, z_b \geq 0.$$

Further details of this approach are given in the appendix.

**Experimental setup** We examine our algorithm under two main conditions: where the true model is linear, and where it is nonlinear. In all cases, we generate problem instances by randomly sampling some $x \in \mathbb{R}^n$ and then generating $p(y|x; \theta)$ according to either $p(y|x; \theta) \propto \exp(\Theta^T x)$ (linear true model) or $p(y|x; \theta) \propto \exp((\Theta^T x)^2)$ (nonlinear true model) for some $\Theta \in \mathbb{R}^{n \times k}$. We compare the following approaches on these tasks: 1) the QP allocation based upon the true model (which performs optimally); 2) MLE approaches (with linear or nonlinear probability models) that fit a model to the data, and then compute the allocation by solving the QP; 3) pure end-to-end policy-optimizing models (using linear or nonlinear hypotheses for the policy); and 4) our task-based learning models (with linear or nonlinear probability models). In all cases, we evaluate test performance by running on 1000 random examples, and evaluate performance over 10 folds of different true $\theta^\star$ parameters.

Figures 2(a) and (b) show the performance of these methods given a linear true model, with linear and nonlinear model hypotheses, respectively. As expected, the linear MLE approach performs best, as the true underlying model is in the class of distributions that it can represent and thus solving the stochastic programming problem is a very strong proxy for solving the true optimization problem under the real distribution. While the true model is also contained within the nonlinear MLE's generic nonlinear distribution class, we see that this method requires more data to converge, and when given less data makes error tradeoffs that are ultimately not the correct tradeoffs for the task at hand; our task-based approach thus outperforms this approach. The task-based approach also substantially outperforms the policy-optimizing neural network, highlighting the fact that it is more data-efficient to run the learning process "through" a reasonable model. Note that here it does not make a difference whether we use the linear or nonlinear model in the task-based approach.

Figures 2(c) and (d) show performance in the case of a nonlinear true model, with linear and nonlinear model hypotheses, respectively. Case (c) represents the "non-realizable" case, where the true underlying distribution cannot be represented by the model hypothesis class. Here, the linear MLE, as expected, performs very poorly: it cannot capture the true underlying distribution, and thus the resultant stochastic programming solution would not be expected to perform well. The linear policy model similarly performs poorly. Importantly, the task-based approach with the *linear* model performs much better here: despite the fact that it still has a misspecified model, the task-based nature of the learning process lets us learn a *different* linear model than the MLE version, which is

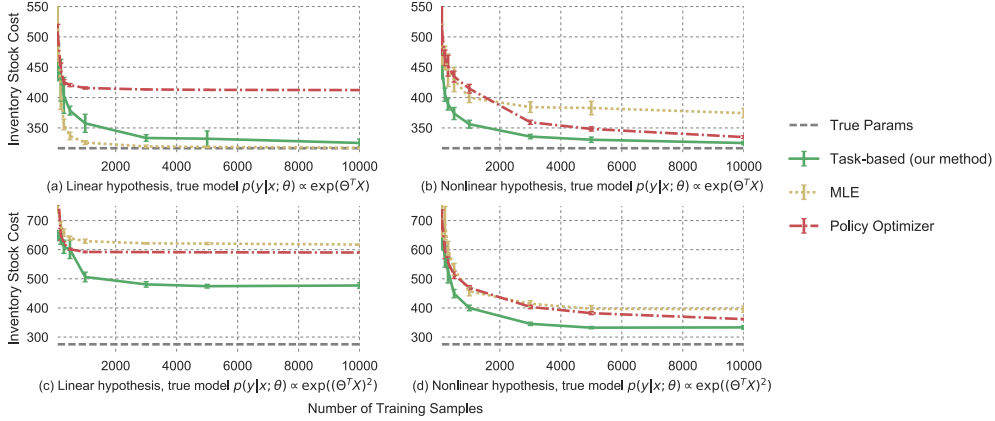

Figure 2: Inventory problem results for 10 runs over a representative instantiation of true parameters ($c_0 = 10, q_0 = 2, c_b = 30, q_b = 14, c_h = 10, q_h = 2$). Cost is evaluated over 1000 testing samples (lower is better). The linear MLE performs best for a true linear model. In all other cases, the task-based models outperform their MLE and policy counterparts.

particularly tuned to the distribution and loss of the task. Finally, also as to be expected, the non-linear models perform better than the linear models in this scenario, but again with the task-based non-linear model outperforming the nonlinear MLE and end-to-end policy approaches.

## 4.2 Load forecasting and generator scheduling

We next consider a more realistic grid-scheduling task, based upon over 8 years of real electrical grid data. In this setting, a power system operator must decide how much electricity generation $z \in \mathbb{R}^{24}$ to schedule for each hour in the next 24 hours based on some (unknown) distribution over electricity demand (Figure 1b). Given a particular realization $y$ of demand, we impose penalties for both generation excess ($\gamma_e$) and generation shortage ($\gamma_s$), with $\gamma_s \gg \gamma_e$. We also add a quadratic regularization term, indicating a preference for generation schedules that closely match demand realizations. Finally, we impose a ramping constraint $c_r$ restricting the change in generation between consecutive timepoints, reflecting physical limitations associated with quick changes in electricity output levels. These are reasonable proxies for the actual economic costs incurred by electrical grid operators when scheduling generation, and can be written as the stochastic programming problem

$$\underset{z \in \mathbb{R}^{24}}{\text{minimize}} \quad \sum_{i=1}^{24} \mathbf{E}_{y \sim p(y|x;\theta)} \left[ \gamma_s [y_i - z_i]_+ + \gamma_e [z_i - y_i]_+ + \frac{1}{2}(z_i - y_i)^2 \right] \tag{11}$$
$$\text{subject to} \quad |z_i - z_{i-1}| \le c_r \ \forall i,$$

where $[v]_+ \equiv \max\{v, 0\}$. Assuming (as we will in our model), that $y_i$ is a Gaussian random variable with mean $\mu_i$ and variance $\sigma_i^2$, then this expectation has a closed form that can be computed via analytically integrating the Gaussian PDF.[4] We then use sequential quadratic programming (SQP) to iteratively approximate the resultant convex objective as a quadratic objective, iterate until convergence, and then compute the necessary Jacobians using the quadratic approximation at the solution, which gives the correct Hessian and gradient terms. Details are given in the appendix.

To develop a predictive model, we make use of a highly-tuned load forecasting methodology. Specifically, we input the past day's electrical load and temperature, the next day's temperature forecast, and additional features such as non-linear functions of the temperatures, binary indicators of weekends or holidays, and yearly sinusoidal features. We then predict the electrical load over all 24

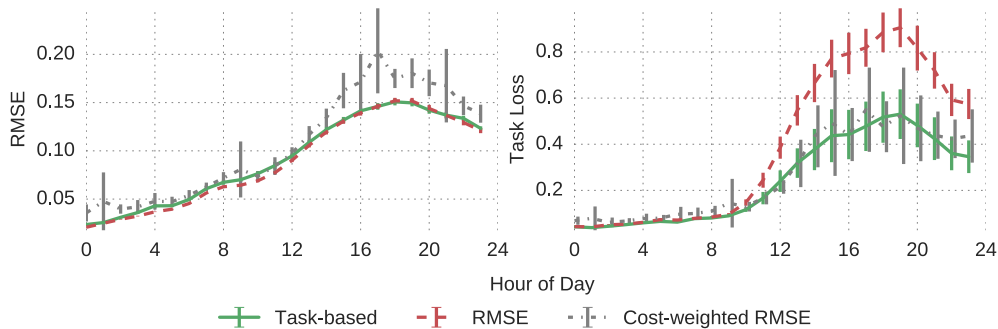

Figure 4: Results for 10 runs of the generation-scheduling problem for representative decision parameters $\gamma_e = 0.5$, $\gamma_s = 50$, and $c_r = 0.4$. (Lower loss is better.) As expected, the RMSE net achieves the lowest RMSE for its predictions. However, the task net outperforms the RMSE net on task loss by 38.6%, and the cost-weighted RMSE on task loss by 8.6%.

hours of the next day. We employ a 2-hidden-layer neural network for this purpose, with an additional residual connection from the inputs to the outputs initialized to the linear regression solution.

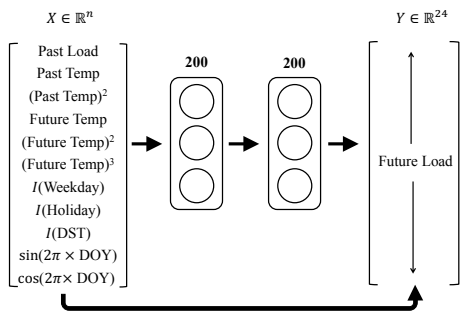

Figure 3: 2-hidden-layer neural network to predict hourly electric load for the next day.

An illustration of the architecture is shown in Figure 3. We train the model to minimize the mean squared error between its predictions and the actual load (giving the mean prediction $\mu_i$), and compute $\sigma_i^2$ as the (constant) empirical variance between the predicted and actual values. In all cases we use 7 years of data to train the model, and 1.75 subsequent years for testing.

Using the (mean and variance) predictions of this base model, we obtain $z^\star(x; \theta)$ by solving the generator scheduling problem (11) and then adjusting network parameters to minimize the resultant task loss. We compare against a traditional stochastic programming model that minimizes just the RMSE, as well as a cost-weighted RMSE that periodically reweights training samples given their task loss.[5] (A pure policy-optimizing network is not shown, as it could not sufficiently learn the ramp constraints. We could not obtain good performance for the policy optimizer even ignoring this infeasibility.)

Figure 4 shows the performance of the three models. As expected, the RMSE model performs best with respect to the RMSE of its predictions (its objective). However, the task-based model substantially outperforms the RMSE model when evaluated on task loss, the actual objective that the system operator cares about: specifically, we improve upon the performance of the traditional stochastic programming method by 38.6%. The cost-weighted RMSE's performance is extremely variable, and overall, the task net improves upon this method by 8.6%.

### 4.3 Price forecasting and battery storage

Finally, we consider a battery arbitrage task, based upon 6 years of real electrical grid data. Here, a grid-scale battery must operate over a 24 hour period based on some (unknown) distribution over future electricity prices (Figure 1c). For each hour, the operator must decide how much to charge ($z_{\text{in}} \in \mathbb{R}^{24}$) or discharge ($z_{\text{out}} \in \mathbb{R}^{24}$) the battery, thus inducing a particular state of charge in the battery ($z_{\text{state}} \in \mathbb{R}^{24}$). Given a particular realization $y$ of prices, the operator optimizes over: 1) profits, 2) flexibility to participate in other markets, by keeping the battery near half its capacity $B$ (with weight $\lambda$), and 3) battery health, by discouraging rapid charging/discharging (with weight $\epsilon$,

| Hyperparameters | | RMSE net | Task-based net (our method) | % Improvement |
|---|---|---|---|---|
| $\lambda$ | $\epsilon$ | | | |
| 0.1 | 0.05 | $-1.45 \pm 4.67$ | $-2.92 \pm 0.30$ | 1.02 |
| 1 | 0.5 | $4.96 \pm 4.85$ | $2.28 \pm 2.99$ | 0.54 |
| 10 | 5 | $131 \pm 145$ | $95.9 \pm 29.8$ | 0.27 |
| 35 | 15 | $173 \pm 7.38$ | $170 \pm 2.16$ | 0.02 |

Table 1: Task loss results for 10 runs each of the battery storage problem, given a lithium-ion battery with attributes $B = 1$, $\gamma_{\text{eff}} = 0.9$, $c_{\text{in}} = 0.5$, and $c_{\text{out}} = 0.2$. (Lower loss is better.) Our task-based net on average somewhat improves upon the RMSE net, and demonstrates more reliable performance.

$\epsilon < \lambda$). The battery also has a charging efficiency ($\gamma_{\text{eff}}$), limits on speed of charge ($c_{\text{in}}$) and discharge ($c_{\text{out}}$), and begins at half charge. This can be written as the stochastic programming problem

$$
\begin{aligned}
\underset{z_{\text{in}}, z_{\text{out}}, z_{\text{state}} \in \mathbb{R}^{24}}{\text{minimize}} \quad & \mathbf{E}_{y \sim p(y|x;\theta)} \left[ \sum_{i=1}^{24} y_i (z_{\text{in}} - z_{\text{out}})_i + \lambda \left\| z_{\text{state}} - \frac{B}{2} \right\|^2 + \epsilon \| z_{\text{in}} \|^2 + \epsilon \| z_{\text{out}} \|^2 \right] \\
\text{subject to} \quad & z_{\text{state},i+1} = z_{\text{state},i} - z_{\text{out},i} + \gamma_{\text{eff}} z_{\text{in},i} \ \forall i, \ \ z_{\text{state},1} = B/2, \\
& 0 \le z_{\text{in}} \le c_{\text{in}}, \ \ 0 \le z_{\text{out}} \le c_{\text{out}}, \ \ 0 \le z_{\text{state}} \le B.
\end{aligned}
\tag{12}
$$

Assuming (as we will in our model) that $y_i$ is a random variable with mean $\mu_i$, then this expectation has a closed form that depends only on the mean. Further details are given in the appendix.

To develop a predictive model for the mean, we use an architecture similar to that described in Section 4.2. In this case, we input the past day's prices and temperature, the next day's load forecasts and temperature forecasts, and additional features such as non-linear functions of the temperatures and temporal features similar to those in Section 4.2. We again train the model to minimize the mean squared error between the model's predictions and the actual prices (giving the mean prediction $\mu_i$), using about 5 years of data to train the model and 1 subsequent year for testing. Using the mean predictions of this base model, we then solve the storage scheduling problem by solving the optimization problem (12), again learning network parameters by minimizing the task loss. We compare against a traditional stochastic programming model that minimizes just the RMSE.

Table 1 shows the performance of the two models. As energy prices are difficult to predict due to numerous outliers and price spikes, the models in this case are not as well-tuned as in our load forecasting experiment; thus, their performance is relatively variable. Even then, in all cases, our task-based model demonstrates better average performance than the RMSE model when evaluated on task loss, the objective most important to the battery operator (although the improvements are not statistically significant). More interestingly, our task-based method shows less (and in some cases, far less) variability in performance than the RMSE-minimizing method. Qualitatively, our task-based method hedges against perverse events such as price spikes that could substantially affect the performance of a battery charging schedule. The task-based method thus yields more reliable performance than a pure RMSE-minimizing method in the case the models are inaccurate due to a high level of stochasticity in the prediction task.

## 5 Conclusions and future work

This paper proposes an end-to-end approach for learning machine learning models that will be used in the loop of a larger process. Specifically, we consider training probabilistic models in the context of stochastic programming to directly capture a task-based objective. Preliminary experiments indicate that our task-based learning model substantially outperforms MLE and policy-optimizing approaches in all but the (rare) case that the MLE model "perfectly" characterizes the underlying distribution. Our method also achieves a 38.6% performance improvement over a highly-optimized real-world stochastic programming algorithm for scheduling electricity generation based on predicted load. In the case of energy price prediction, where there is a high degree of inherent stochasticity in the problem, our method demonstrates more reliable task performance than a traditional predictive method. The task-based approach thus demonstrates promise in optimizing in-the-loop predictions. Future work includes an extension of our approach to stochastic learning models with multiple rounds, and further to model predictive control and full reinforcement learning settings.

## Acknowledgments

This material is based upon work supported by the National Science Foundation Graduate Research Fellowship Program under Grant No. DGE1252522, and by the Department of Energy Computational Science Graduate Fellowship.

## Footnotes

[1]It is standard to presume in stochastic programming that equality constraints depend only on decision variables (not random variables), as non-trivial random equality constraints are typically not possible to satisfy.

[2]This distinction is roughly analogous to the policy search vs. model-based settings in reinforcement learning. However, for the purposes of this paper, we consider much simpler stochastic programs without the multiple rounds that occur in RL, and the extension of these techniques to a full RL setting remains as future work.

[3]This is referred to as a two-stage stochastic programming problem (though a very trivial example of one), where first stage variables consist of the amount of product to buy before observing demand, and second-stage variables consist of how much to sell back or additionally purchase once the true demand has been revealed.

[4] Part of the philosophy behind applying this approach here is that we *know* the Gaussian assumption is incorrect: the true underlying load is neither Gaussian distributed nor homoskedastic. However, these assumptions are exceedingly common in practice, as they enable easy model learning and exact analytical solutions. Thus, training the (still Gaussian) system with a task-based loss retains computational tractability while still allowing us to modify the distribution's parameters to improve actual performance on the task at hand.

[5]It is worth noting that a cost-weighted RMSE approach is only possible when direct costs can be assigned independently to each decision point, i.e. when costs do not depend on multiple decision points (as in this experiment). Our task-based method, however, accommodates the (typical) more general setting.

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
