[Supplementary Material]

# A  Appendix

We present some computational and architectural details for the proposed task-based learning model, both in the general case and for the experiments described in Section 4.

## A.1  Differentiating the optimization solution to a stochastic programming problem

The issue of chief technical challenge to our approach is computing the gradient of an objective that depends upon the argmin operation $z^\star(x; \theta)$. Specifically, we need to compute the term

$$\frac{\partial L}{\partial \theta} = \frac{\partial L}{\partial z^\star} \frac{\partial z^\star}{\partial \theta} \tag{A.1}$$

which involves the Jacobian $\frac{\partial z^\star}{\partial \theta}$. This is the Jacobian of the optimal solution with respect to the distribution parameters $\theta$. Recent approaches have looked into similar argmin differentiations [28, 29], though the methodology we present here is more general and handles the stochasticity of the objective.

We begin by writing the KKT optimality conditions of the general stochastic programming problem (3), where all expectations are taken with respect to the modeled distribution $y \sim p(y|x; \theta)$ (for compactness, denoted here as $\mathbf{E}_{y_\theta}$). Further, assuming the problem is convex means we can replace the general equality constraints $h(z) = 0$ with the linear constraint $Az = b$. A point $(z, \lambda, \nu)$ is a primal-dual optimal point if it satisfies

$$
\begin{aligned}
\mathbf{E}_{y_\theta} g(z) &\leq 0 \\
Az &= b \\
\lambda &\geq 0 \\
\lambda \circ \mathbf{E}_{y_\theta} g(z) &= 0 \\
\nabla_z \mathbf{E}_{y_\theta} f(z) + \lambda^T \nabla_z \mathbf{E}_{y_\theta} g(z) + A^T \nu &= 0
\end{aligned}
\tag{A.2}
$$

where here $g$ denotes the vector of all inequality constraints (represented as a vector-valued function), and where we wrap the dependence on $x$ and $y$ into the functions $f$ and $g_i$ themselves.

Differentiating these equations and applying the implicit function theorem gives a set of linear equations that we can solve to obtain the necessary Jacobians

$$
\begin{bmatrix}
\nabla_z^2 \mathbf{E}_{y_\theta} f(z) + \sum_{i=1}^{n_{ineq}} \lambda_i \nabla_z^2 \mathbf{E}_{y_\theta} g_i(z) & (\nabla_z \mathbf{E}_{y_\theta} g(z))^T & A^T \\
\operatorname{diag}(\lambda) \left( \nabla_z \mathbf{E}_{y_\theta} g(z) \right) & \operatorname{diag}(\mathbf{E}_{y_\theta} g(z)) & 0 \\
A & 0 & 0
\end{bmatrix}
\begin{bmatrix}
\frac{\partial z}{\partial \theta} \\
\frac{\partial \lambda}{\partial \theta} \\
\frac{\partial \nu}{\partial \theta}
\end{bmatrix}
=
\begin{bmatrix}
\frac{\partial \nabla_z \mathbf{E}_{y_\theta} f(z)}{\partial \theta} + \frac{\partial \sum_{i=1}^{n_{ineq}} \lambda_i \nabla_z \mathbf{E}_{y_\theta} g_i(z)}{\partial \theta} \\
\operatorname{diag}(\lambda) \frac{\partial \mathbf{E}_{y_\theta} g(z)}{\partial \theta} \\
0
\end{bmatrix}.
\tag{A.3}
$$

The terms on the left side are the optimality conditions of the convex problem, and the terms on right side are the derivatives of the relevant functions at the achieved solution, with respect to the governing parameter $\theta$. These equations will take slightly different forms depending on how the stochastic programming problem is solved, but are usually fairly straightforward to compute if the solution is solved in some "exact" manner (i.e., where second order information is used). In practice, we calculate the right side of this equation by employing sequential quadratic programming [30] to find the optimal policy $z^\star$ for the given parameters $\theta$, using a recently-proposed approach for fast solution of argmin differentiation for QPs [31] to solve the necessary linear equations; we then take the derivatives at the optimum produced by this strategy.

## A.2  Details on computation for inventory stock problem

The objective for our "conditional" variation of the classical inventory stock problem is

$$f_{stock}(y, z) = c_0 z + \frac{1}{2} q_0 z^2 + c_b [y - z]_+ + \frac{1}{2} q_b ([y - z]_+)^2 + c_h [z - y]_+ + \frac{1}{2} q_h ([z - y]_+)^2 \tag{A.4}$$

where $z$ is the amount of product ordered; $y$ is the stochastic electricity demand (which is affected by features $x$); $[v]_+ \equiv \max\{v, 0\}$; and $(c_0, q_0)$, $(c_b, q_b)$, and $(c_h, q_h)$ are linear and quadratic costs on the amount of product ordered, over-orders, and under-orders, respectively. Our proxy stochastic programming problem can then be written as

$$\underset{z}{\text{minimize}} \quad L(\theta) = \mathbf{E}_{y \sim p(y|x; \theta)}[f_{stock}(y, z)]. \tag{A.5}$$

To simplify the setting, we further assume that the demands are discrete, taking on values $d_1, \ldots, d_k$ with probabilities (conditional on $x$) $(p_\theta)_i \equiv p(y = d_i|x; \theta)$. Thus our stochastic programming problem (A.5) can be written succinctly as a joint quadratic program

$$\underset{z \in \mathbb{R}, z_b, z_h \in \mathbb{R}^k}{\text{minimize}} \quad c_0 z + \frac{1}{2} q_0 z^2 + \sum_{i=1}^{k} (p_\theta)_i \left( c_b(z_b)_i + \frac{1}{2} q_b(z_b)_i^2 + c_h(z_h)_i + \frac{1}{2} q_h(z_h)_i^2 \right) \tag{A.6}$$

$$\text{subject to} \quad d - z\mathbf{1} \leq z_b, \quad z\mathbf{1} - d \leq z_h, \quad z, z_h, z_b \geq 0.$$

To demonstrate the explicit formula for argmin operation Jacobians for this particular case (e.g., to compute the terms in (A.3)), note that we can write the above QP in inequality form as $\text{minimize}_{\{z:Gz \leq h\}} \frac{1}{2} z^T Q z + c^T z$ with

$$z = \begin{bmatrix} z \\ z_b \\ z_h \end{bmatrix}, \; Q = \begin{bmatrix} q_0 & 0 & 0 \\ 0 & q_b p_\theta & 0 \\ 0 & 0 & q_h p_\theta \end{bmatrix}, \; c = \begin{bmatrix} c_0 \\ c_b p_\theta \\ c_h p_\theta \end{bmatrix}, \; G = \begin{bmatrix} -1 & -I & 0 \\ 1 & 0 & -I \\ -1 & 0 & 0 \\ 0 & -I & 0 \\ 0 & 0 & -I \end{bmatrix}, \; h = \begin{bmatrix} -d \\ d \\ 0 \\ 0 \\ 0 \end{bmatrix}.$$
$$\tag{A.7}$$

Thus, for an optimal primal-dual solution $(z^\star, \lambda^\star)$, we can compute the Jacobian $\frac{\partial z^\star}{\partial p_\theta}$ (the Jacobian of the optimal solution with respect to the probability vector $p_\theta$ mentioned above), via the formula

$$\begin{bmatrix} \frac{\partial z^\star}{\partial p_\theta} \\ \frac{\partial \lambda^\star}{\partial p_\theta} \end{bmatrix} = \begin{bmatrix} Q & G^T \\ D(\lambda^\star)G & D(Gz^\star - h) \end{bmatrix}^{-1} \begin{bmatrix} 0 \\ q_b z_b^\star + c_b \mathbf{1} \\ q_h z_h^\star + c_h \mathbf{1} \\ 0 \end{bmatrix}, \tag{A.8}$$

where $D(\cdot)$ denotes a diagonal matrix for an input vector. After solving the problem and computing these Jacobians, we can compute the overall gradient with respect to the task loss $L(\theta)$ via the chain rule

$$\frac{\partial L}{\partial \theta} = \frac{\partial L}{\partial z^\star} \frac{\partial z^\star}{\partial p_\theta} \frac{\partial p_\theta}{\partial \theta} \tag{A.9}$$

where $\frac{\partial p_\theta}{\partial \theta}$ denotes the Jacobian of the model probabilities with respect to its parameters, which are computed in the typical manner. Note that in practice, these Jacobians need not be computed explicitly, but can be computed efficiently via backpropagation; we use a recently-developed differentiable batch QP solver [31] to both solve the optimization problem in QP form and compute its derivatives.

## A.3 Details on computation for power scheduling problem

The objective for the load forecasting problem is given by

$$\underset{z \in \mathbb{R}^{24}}{\text{minimize}} \quad \sum_{i=1}^{24} \mathbf{E}_{y \sim p(y|x;\theta)} \left[ \gamma_s [y_i - z_i]_+ + \gamma_e [z_i - y_i]_+ + \frac{1}{2}(z_i - y_i)^2 \right] \tag{A.10}$$

$$\text{subject to} \quad |z_i - z_{i-1}| \leq c_r \; \forall i,$$

where $z$ is the generator schedule, $y$ is the stochastic demand (which is affected by features $x$), $[v]_+ \equiv \max\{v, 0\}$, $\gamma_e$ is an over-generation penalty, $\gamma_s$ is an under-generation penalty, and $c_r$ is a ramping constraint. Assuming that $y_i$ is a Gaussian random variable with mean $\mu_i$ and variance $\sigma_i^2$, then this expectation has a closed form that can be computed via analytically integrating the Gaussian PDF. Specifically, this closed form is

$$\mathbf{E}_{y \sim p(y|x;\theta)} \left[ \gamma_s [y_i - z_i]_+ + \gamma_e [z_i - y_i]_+ + \frac{1}{2}(z_i - y_i)^2 \right]$$

$$= \underbrace{(\gamma_s + \gamma_e)(\sigma^2 p(z_i; \mu, \sigma^2) + (z_i - \mu)F(z_i; \mu, \sigma^2)) - \gamma_s(z_i - \mu)}_{\alpha(z_i)} + \frac{1}{2}((z_i - \mu_i)^2 + \sigma_i^2),$$
$$\tag{A.11}$$

where $p(z; \mu, \sigma^2)$ and $F(z; \mu, \sigma^2)$ denote the Gaussian PDF and CDF, respectively with the given mean and variance. This is a convex function of $z$ (not apparent in this form, but readily established

because it is an expectation of a convex function), and we can thus optimize it efficiently and compute the necessary Jacobians.

Specifically, we use sequential quadratic programming (SQP) to iteratively approximate the resultant convex objective as a quadratic objective, and iterate until convergence; specifically, we repeatedly solve

$$z^{(k+1)} = \underset{z}{\operatorname{argmin}} \ \frac{1}{2} z^T \operatorname{diag}\left(\frac{\partial^2 \alpha(z_i^{(k)})}{\partial z^2} + 1\right) z + \left(\frac{\partial \alpha(z^{(k)})}{\partial z} - \mu\right)^T z \tag{A.12}$$

$$\text{subject to } |z_i - z_{i-1}| \le c_r \ \forall i$$

until $||z^{(k+1)} - z^{(k)}|| < \delta$ for a small $\delta$, where

$$\begin{aligned}
\frac{\partial \alpha}{\partial z} &= (\gamma_s + \gamma_e) F(z; \mu, \sigma) - \gamma_s, \\
\frac{\partial^2 \alpha}{\partial z^2} &= (\gamma_s + \gamma_e) p(z; \mu, \sigma).
\end{aligned} \tag{A.13}$$

We then compute the necessary Jacobians using the quadratic approximation (A.12) at the solution, which gives the correct Hessian and gradient terms. We can furthermore differentiate the gradient and Hessian with respect to the underlying model parameters $\mu$ and $\sigma^2$, again using a recently-developed batch QP solver [31].

## A.4  Details on computation for battery storage problem

The objective for the battery storage problem is given by

$$\underset{z_{\mathrm{in}}, z_{\mathrm{out}}, z_{\mathrm{state}} \in \mathbb{R}^{24}}{\operatorname{minimize}} \ \mathbf{E}_{y \sim p(y|x;\theta)}\left[\sum_{i=1}^{24} y_i (z_{\mathrm{in}} - z_{\mathrm{out}})_i + \lambda \left\|z_{\mathrm{state}} - \frac{B}{2}\right\|^2 + \epsilon\|z_{\mathrm{in}}\|^2 + \epsilon\|z_{\mathrm{out}}\|^2\right] \tag{A.14}$$

$$\text{subject to } z_{\mathrm{state},i+1} = z_{\mathrm{state},i} - z_{\mathrm{out},i} + \gamma_{\mathrm{eff}} z_{\mathrm{in},i} \ \forall i, \ z_{\mathrm{state},1} = B/2,$$

$$0 \le z_{\mathrm{in}} \le c_{\mathrm{in}}, \ 0 \le z_{\mathrm{out}} \le c_{\mathrm{out}}, \ 0 \le z_{\mathrm{state}} \le B,$$

where $z_{\mathrm{in}}, z_{\mathrm{out}}, z_{\mathrm{state}}$ are decisions over the charge amount, discharge amount, and resultant state of the battery, respectively; $y$ is the stochastic electricity price (which is affected by features $x$); $B$ is the battery capacity; $\gamma_{\mathrm{eff}}$ is the battery charging efficiency; $c_{\mathrm{in}}$ and $c_{\mathrm{out}}$ are maximum hourly charge and discharge amounts, respectively; and $\lambda$ and $\epsilon$ are hyperparameters related to flexibility and battery health, respectively.

Assuming $y_i$ is a random variable with mean $\mu_i$, the expectation in the objective has a closed form:

$$\mathbf{E}_{y \sim p(y|x;\theta)}\left[\sum_{i=1}^{24} y_i (z_{\mathrm{in}} - z_{\mathrm{out}})_i + \lambda \left\|z_{\mathrm{state}} - \frac{B}{2}\right\|^2 + \epsilon\|z_{\mathrm{in}}\|^2 + \epsilon\|z_{\mathrm{out}}\|^2\right]$$

$$= \sum_{i=1}^{24} \mu_i (z_{\mathrm{in}} - z_{\mathrm{out}})_i + \lambda \left\|z_{\mathrm{state}} - \frac{B}{2}\right\|^2 + \epsilon\|z_{\mathrm{in}}\|^2 + \epsilon\|z_{\mathrm{out}}\|^2. \tag{A.15}$$

We can then write this expression in QP form as $\operatorname{minimize}_{\{z: Gz \le h, \ Az=b\}} \frac{1}{2} z^T Q z + c^T z$ with

$$\boldsymbol{z} = \begin{bmatrix} z_{\mathrm{in}} \\ z_{\mathrm{out}} \\ z_{\mathrm{state}} \end{bmatrix}, \ Q = \begin{bmatrix} \epsilon I & 0 & 0 \\ 0 & \epsilon I & 0 \\ 0 & 0 & \lambda I \end{bmatrix}, \ c = \begin{bmatrix} \mu \\ -\mu \\ -\lambda B \mathbf{1} \end{bmatrix},$$

$$G = \begin{bmatrix} I & 0 & 0 \\ -I & 0 & 0 \\ 0 & I & 0 \\ 0 & -I & 0 \\ 0 & 0 & I \\ 0 & 0 & -I \end{bmatrix}, \ h = \begin{bmatrix} c_{\mathrm{in}} \\ 0 \\ c_{\mathrm{out}} \\ 0 \\ B \\ 0 \end{bmatrix}, \ A = \begin{bmatrix} 0 & 0 & 0,\ldots,0,1 \\ \gamma_{\mathrm{eff}} D_1^T & -D_1^T & D_1^T - D_2^T \end{bmatrix} \ b = \begin{bmatrix} B/2 \\ 0 \end{bmatrix},$$

(A.16)

where $D_1 = \begin{bmatrix} I \\ 0 \end{bmatrix} \in \mathbb{R}^{24 \times 23}$ and $D_2 = \begin{bmatrix} 0 \\ I \end{bmatrix} \in \mathbb{R}^{24 \times 23}$.

For this experiment, we assume that $y_i$ is a lognormal random variable (with mean $\mu_i$); thus, to obtain our predictions, we predict the mean of $\log(y)$ (i.e., we predict $\log(\mu)$). After obtaining these predictions, we solve (A.4), compute the necessary Jacobians at the solution, and update the underlying model parameter $\mu$ via backpropagation, again using [31].

## A.5  Implementation notes

For all linear models, we use a one-layer linear neural network with the appropriate input and output layer dimensions. For all nonlinear models, we use a two-hidden-layer neural network, where each "layer" is actually a combination of linear, batch norm [32], ReLU, and dropout ($p = 0.2$) layers with dimension 200. In both cases, we add an additional softmax layer in cases where probability distributions are being predicted.

All models are implemented using PyTorch[A.1] and employ the Adam optimizer [33]. All QPs are solved using a recently-developed differentiable batch QP solver [31], and Jacobians are also computed automatically using backpropagation via the same.

Source code for all experiments is available at `https://github.com/locuslab/e2e-model-learning`.

---

[A.1]`https://pytorch.org`

## Acknowledgments

This material is based upon work supported by the National Science Foundation Graduate Research Fellowship Program under Grant No. DGE1252522, and by the Department of Energy Computational Science Graduate Fellowship.