[Reviews · NeurIPS 2017]

Reviewer 1



The paper proposes an approach to train predictive models performance of which is not based on classic likelihood objectives, but is instead based on performance on some task external to the model. In order to achieve this, the model parameters are optimized so as to minimize the loss on external task, which in turn may involve a sub-optimization problem that depends on model parameters. A synthetic and real-data experiments are presented that clearly illustrate the usefulness of proposed approach. The introduction is very well-motivated and the exposition is generally clear. The paper is technically sound and builds on sound foundations - I see no obvious flaws. The paper contains good motivation about why the proposed approach is necessary. I see this work as a worthwhile and well-motivated application of existing technical contributions. The important technical piece that make this approach possible, which is differentiation though an argmax are already presented in [Amos 2016]. While building on existing results, this work applies them in a very relevant and well-motivated formulation of task-based learning and I believe would be of interest to the machine learning community. The proposed benefit, but also a potential issue of the proposed approach is that the model is not independent of the task. Is is possible to characterize how much the model may be overfitting to the specific task and whether it generalizes well to a different (potentially only slightly different) task? It would be good to reference related work on meta-learning “Model-Agnostic Meta-Learning for Fast Adaptation of Deep Networks” by Finn et al, as that’s another example of differentiation through an optimization procedure.

Reviewer 2



--Brief summary of the paper: The paper proposes a learning method for solving two-stage stochastic programming problems which involve minimizing f(x,y,z) w.r.t. z. The main idea of the paper is to learn a predictive model p(y|x;theta) such that the task's objective function f is directly optimized. In contrast, traditional approaches learn p(y|x;theta) to minimize a prediction error without considering f. The main technical challenge in the paper is to solve a sub-optimization problem involving argmin w.r.t. z, and the proposed method can do so in an efficient manner by assuming that the optimization problem is convex in z. The method is experimentally evaluated on two problems and it is shown to outperform traditional methods. --Major comments: The idea of adopting end-to-end learning to solve two-stage stochastic programming is interesting. However, I have a major concern for the proposed method which is the lack of convergence guarantees. Since the optimization problem is assumed to be convex in z, the obtained solution z*(x;theta) is supposed to be the "true" optimal if data is drawn from the true distribution p(x,y). However, a solution obtained using the predictive model p(y|x;theta) is unlikely to be true optimal unless p(y|x;theta) is the true conditional distribution p(y|x). (This issue is commonly known as model bias in the context of model-based reinforcement learning which usually involves non-convex objectives.) Since the proposed method does not theoretically guarantee that p(y|x;theta) converges to p(y|x) even when the model hypothesis is correct, it seems likely that even for a convex optimization problem the method may only find a sub-optimal solution. For this reason, I think having convergence guarantees or error bounds either for the predictive model or for the obtained solution itself are very important to theoretically justify the method and would be a significant contribution to the paper. --Questions: 1) It is not clear why Algorithm 1 requires mini-batches training since Line 7 of the algorithm only checks the constraint for a single sample. 2) In the first experiment, why does the performance of the end-to-end policy optimization method depend on the model hypothesis when it does not rely on a predictive model? --Minor suggestions: 1) In line 154 the paper argue that the model-free approach requires a rich policy class and is data inefficient. However, the model-based approach also requires a rich model class as well. Moreover, the model-based approach can suffer from model bias while the model-free approach cannot. 2) The applicability of the proposed method is quite limited. As mentioned in the paper, solving a sub-optimization problem with argmin is not trivial and convexity assumption can help in this regard. However, practical decision making problems may involve non-convex or unknown objective functions. A variant of the proposed method that is applicable to these tasks would make the method more appealing. 3) The last term of Eq.(4) should have an expectation over the density of x. --Comments after author's response: I feel more positive about the paper after reading the author’s response. Now I think that the proposed method is an important contribution to the field and I will increase my score. However, I am still not convince that the proposed method will be useful outside domains with convex objectives without empirical evidences.

Reviewer 3



Frankly I didn't understand the problem setup. What are \mathcal{X, Y, Z}? compact sets in R^n? Unbounded sets? Or maybe these are finite-sets? I understand that they describe an end-to-end optimization scheme like [29], but the problem set of stochastic programming (e.g. inventory stacking ) is unfamiliar to me. The results of the load forecasting task looks like it might be impressive, but again, I am unfamiliar with these problems. I'd love to see result on problems more familiar to me (and I suspect the NIPS crowd in general) e.g. continuous control problems.